# Oncogenic Role of Secreted Engrailed Homeobox 2 (EN2) in Prostate Cancer

**DOI:** 10.3390/jcm8091400

**Published:** 2019-09-06

**Authors:** Enrique Gómez-Gómez, Juan M. Jiménez-Vacas, Sergio Pedraza-Arévalo, Fernando López-López, Vicente Herrero-Aguayo, Daniel Hormaechea-Agulla, José Valero-Rosa, Alejandro Ibáñez-Costa, Antonio J. León-González, Rafael Sánchez-Sánchez, Teresa González-Serrano, Maria J. Requena-Tapia, Justo P. Castaño, Julia Carrasco-Valiente, Manuel D. Gahete, Raúl M. Luque

**Affiliations:** 1Maimonides Institute for Biomedical Research of Cordoba (IMIBIC), 14004 Córdoba, Spain; enrique.gomez.gomez.sspa@juntadeandalucia.es (E.G.-G.); b12jivaj@uco.es (J.M.J.-V.); b92pears@uco.es (S.P.-A.); ferll_@hotmail.com (F.L.-L.); b22heagv@uco.es (V.H.-A.); hormaechea85@gmail.com (D.H.-A.); jose.valero.rosa.sspa@juntadeandalucia.es (J.V.-R.); aibanezcosta@gmail.com (A.I.-C.); antonio.leon@imibic.org (A.J.L.-G.); patologiahrs@gmail.com (R.S.-S.); mariat.gonzalez.serrano.sspa@juntadeandalucia.es (T.G.-S.); bc1cafuj@uco.es (J.P.C.); julia.carrasco.sspa@juntadeandalucia.es (J.C.-V.); bc2gaorm@uco.es (M.D.G.); 2Department of Cell Biology, Physiology, and Immunology, Universidad de Córdoba, 14004 Córdoba, Spain; 3Reina Sofia University Hospital, 14004 Córdoba, Spain; 4Urology Service, Reina Sofia University Hospital, 14004 Córdoba, Spain; 5CIBER Fisiopatología de la Obesidad y Nutrición (CIBERobn), 14004 Córdoba, Spain; 6Anatomical Pathology Service, Reina Sofia University Hospital, 14004 Córdoba, Spain

**Keywords:** engrailed homeobox variants, prostate cancer, aggressiveness, biomarker

## Abstract

Engrailed variant-2 (EN2) has been suggested as a potential diagnostic biomarker; however, its presence and functional role in prostate cancer (PCa) cells is still controversial or unknown. Here, we analyzed 1) the expression/secretion profile of EN2 in five independent samples cohorts from PCa patients and controls (prostate tissues and/or urine) to determine its utility as a PCa biomarker; and 2) the functional role of EN2 in normal (RWPE1) and tumor (LNCaP/22Rv1/PC3) prostate cells to explore its potential value as therapeutic target. EN2 was overexpressed in our two cohorts of PCa tissues compared to control and in tumor cell lines compared with normal-like prostate cells. This profile was corroborated in silico in three independent data sets [The Cancer Genome Atlas(TCGA)/Memorial Sloan Kettering Cancer Center (MSKCC)/Grasso]. Consistently, urine EN2 levels were elevated and enabled discrimination between PCa and control patients. EN2 treatment increased cell proliferation in LNCaP/22Rv1/PC3 cells, migration in RWPE1/PC3 cells, and PSA secretion in LNCaP cells. These effects were associated, at least in the androgen-sensitive LNCaP cells, with increased AKT and androgen-receptor phosphorylation levels and with modulation of key cancer-associated genes. Consistently, EN2 treatment also regulated androgen-receptor activity (full-length and splicing variants) in androgen-sensitive 22Rv1 cells. Altogether, this study demonstrates the potential utility of EN2 as a non-invasive diagnostic biomarker for PCa and provides novel and valuable information to further investigate its putative utility to develop new therapeutic tools in PCa.

## 1. Introduction

Prostate cancer (PCa) is diagnosed in approximately 899,000 men per year worldwide [1] and is the most frequent non-skin cancer in developed countries among men [2]. Since the 1990s, with the introduction of the prostate specific antigen (PSA) test for the detection of PCa, the possibility of early diagnosis has been improved and, consequently, metastatic disease and specific mortality rates have been reduced in most Western countries [3]. However, the management of patients with PCa still faces several limitations. Firstly, the PSA test displays low specificity due to the influence of multiple factors that increase PSA levels, such as benign prostatic hyperplasia or prostatitis [4], and it is not able to accurately distinguish clinically-relevant tumors from indolent cases. Similarly, various PSA-related derivatives, such as PSA velocity, PSA density, and free-to-total PSA ratio, have only provided limited improvements in terms of specificity [5]. Thus, there is no consensus-based recommendation with regard to population screening based on PSA measurement due to the proven risk of over-diagnosis and over-treatment in a considerable number of patients [6]. Secondly, progression of PCa is tremendously complex and its treatment is severely hampered by the lack of satisfactory therapeutic alternatives. Indeed, a significant number of patients are resistant or develop resistance to hormonal castration, the first-line medical therapy in this cancer type, and their disease progresses towards a castration-resistant state (CRPC), wherein the therapeutic alternatives are limited and, in many cases, insufficient [7,8]. Therefore, it seems essential to validate alternative diagnostic biomarkers to complement the PSA test, and to identify novel molecular targets in order to develop additional and more effective therapeutic tools.

In this scenario, the homeodomain-containing transcription factors comprise a gene family that controls cell and tissue identity during normal embryonic development, and have been shown to be strikingly re-expressed by different tumor cell types [9], wherein they could provide novel diagnostic biomarkers or therapeutic targets. Together with *HOX* and *PAX*, *engrailed-homeobox* (*EN*) genes are key members of the homeobox family. *EN* genes were originally characterized in *Drosophila melanogaster* and, later on, in different vertebrates species [10,11]. In human, two *EN* genes (*EN1* and *EN2*) located on chromosome 2 (2q14.2) and 7 (7q36.3), respectively, have been discovered, which slightly differ in function [12]. EN proteins are transcription factors capable to modulate multiple processes at different stages of development, involving transcriptional and translational regulation [9], but also present an unconventional ability to be secreted from producing cells, and to be internalized by others [13,14]. The main role of these proteins during embryonic development is to regulate neural development and embryonic axonal guidance [15]. However, they have been also shown to be expressed in different tumor pathologies, such as leukemia, glioblastoma, colon, ovarian, breast, bladder, and PCa [16,17,18]. In the particular case of PCa, EN2 has been found to be over-expressed in human PCa cells compared to normal prostate epithelial cells or stroma cells [19,20], suggesting its putative utility as a PCa biomarker. In fact, some studies have shown that EN2 can be detected in urine from PCa patients, wherein it could serve as a non-invasive diagnostic biomarker [20,21,22,23,24]. However, the accuracy of EN2 as diagnostic biomarker and the methodological procedure for EN2 assessment in urine samples are still a matter of debate, inasmuch as the sensitivity and specificity of this biomarker is considerably variable among studies and the values fluctuate depending on the existence of previous prostate massage [20,21,22,23,24]. Moreover, little is known about the potential tumorigenic role of EN2 in PCa since only a single study has shown that its silencing could be associated to a decrease in PCa cell proliferation [19]. Strikingly, EN2 protein does not seem to be localized in the nuclei of PCa cells but, rather, close to the luminal border of the cells, associated to secretory blebs [20]. Accordingly, it has been reported that different established PCa cell lines can release EN2 protein to the media, thereby suggesting that secreted EN2 could play a pathological role in PCa [20]. However, this pathological role has been poorly explored hitherto, and, consequently, it is not known if EN2 could provide novel therapeutic targets for this highly incident and prevalent pathology.

Therefore, based on the information mentioned above, the objectives of this study were: (1) To analyze the utility of EN2 as a non-invasive diagnostic biomarker by measuring its expression and secretion levels in different, independent cohorts of samples from PCa patients and controls (prostate tissues and urine); and (2) to investigate the oncogenic role of EN2 and its underlying molecular mechanisms as well as its putative value as a therapeutic target in PCa by using different prostate cell lines (normal (RWPE-1) and tumor (LNCaP, 22Rv1 and PC3) cells) and diverse experimental approaches.

## 2. Material and Methods

### 2.1. Patients and Samples

This study was approved by the Ethics Committee of the IMIBIC/Reina Sofia University Hospital (Córdoba, Spain), performed according to the Declaration of Helsinki, and patients were treated following national and international clinical practice guidelines. A written informed consent was required before collection of samples, which were managed by the Andalusian Health System Biobank (Córdoba, Spain). The study of tissue samples included: (1) Formalin-fixed paraffin-embedded (FFPE) PCa tissues (*n* = 33) obtained from radical prostatectomies of patients diagnosed with clinically localized low-intermediate grade PCa (Table 1), which presented tumor and adjacent non-tumor control tissues, and; (2) fresh biopsy samples (*n* = 23) from patients with locally-advanced PCa (palpable in digital rectal examination (DRE)) and fresh non-tumor prostate samples (NPs, *n* = 7) derived from patients that underwent cystoprostatectomy due to bladder cancer (Table 2). All diagnoses (tumor and non-tumor cases) were confirmed by specialist uropathologists. Evaluations of prostatectomies and biopsies were performed following the 2010 and modified 2005 ISUP criteria, respectively [25,26]. Urine samples were collected between 8:00 and 10:00 after, at least, eight hours of fasting in 1.5 mL aliquots and stored at −80 °C for subsequent analyses. Urine samples were obtained from: (1) Patients with PCa confirmed by positive biopsy (*n* = 24), and (2) control individuals, which included, first, subjects with no suspicious urologic symptoms, low PSA (<2.5 ng/mL) and normal DRE who voluntarily participated in this study (*n* = 10), and, second, patients with suspect of PCa but negative results on the systematic trans-rectal ultrasound-guided biopsy, which showed PSA < 10 ng/mL (*n* = 10; Table 3). It should be mentioned that urine levels of EN2 after DRE were also analyzed in the cohort of patients with PCa (*n* = 24).

### 2.2. Datasets Analysis

Processed freely available RNA-seq data from The Cancer Genome Atlas (TCGA, https://gdc-portal.nci.nih.gov/) and the Memorial Sloan Kettering Cancer Center (MSKCC, https://www.mskcc.org/) regarding Prostate Cancer Adenocarcinoma (PRAD) were compiled and used for subsequent analysis. In addition, available PCa Grasso cohort data from Gen expression Omnibus (GSE35988) were also used for the analysis. Furthermore, free available cell lines data from Cancer Cell Line Encyclopedia (https://portals.broadinstitute.org/ccle) was also used.

### 2.3. EN2 Protein, Reagents, and Cell Lines

Recombinant protein of EN2 was purchased from Origene (TP311220; Origene, Rockville, MD, USA) and IGF1 and Paclitaxel from Life Technologies (Madrid, Spain).

Normal-like prostate cell line (RWPE-1) and PCa cell lines (LNCaP, 22Rv1 and PC3) were obtained from ATCC, validated by analysis of Short Tandem Repeats (STRs) (GenePrint^®^ 10 System, Promega, Barcelona, Spain), and checked for mycoplasma contamination by PCR, as previously reported [27]. RWPE-1 cells were cultured in Keratinocyte-serum free medium (SFM)(Gibco, Waltham, MA, USA), while LNCaP, 22Rv1 and PC3 cells were maintained in Roswell Park Memorial Institute medium (RPMI) 1640 (Lonza, Basel, Switzerland), supplemented with 10% Fetal Bovine Serum (FBS), 1% glutamine, and 0.2% antibiotic, as previously reported [27,28]. All cell lines were grown at 37 °C, in a humidified atmosphere with 5.0% CO_2_.

### 2.4. RNA Isolation, Reverse-Transcription, and Real-Time Quantitative PCR (qPCR)

Total RNA from FFPE samples was isolated using the RNeasy FFPE Kit (Qiagen, Limburg, Netherlands) following the manufacturer’s instructions. The set of fresh samples was extracted using the AllPrep DNA/RNA/Protein Mini Kit followed by deoxyribonuclease treatment using RNase-Free DNase Set (Qiagen, Limburg, Netherlands). Total RNA was also isolated from cell lines using TRIzol Reagent (Life Technologies, Barcelona, Spain) following the manufacturer’s protocol and subsequently treated with DNase (Promega, Barcelona, Spain). Quantification of the recovered RNA was assessed using NanoDrop 2000 spectrophotometer (Thermo Scientific, Wilmington, NC, USA). Briefly, one microgram of total RNA was retro-transcribed to cDNA with the First Strand Synthesis kit using random hexamer primers (Thermo Scientific, Madrid, Spain). cDNAs were amplified with the Brilliant III SYBR Green Master Mix (Stratagene, La Jolla, CA, USA) using the Stratagene Mx3000p system and specific primers for each transcript of interest. Expression levels (absolute mRNA copy number/50 ng of sample) of *EN1* (sense: GCAACCCGGCTATCCTACTT; antisense: CGATCCGAATAACGTGTGC) and *EN2* (sense: GAACCCGAACAAAGAGGACA; antisense: ACCTGTTGGTCTGGAACTCG) were measured using primers designed with Primer3 software and methods previously reported [29,30]. Normalization of all genes was done according to *GAPDH* expression levels or to a normalization factor, obtained by the expression levels of two control genes (*GAPDH*, sense: AATCCCATCACCATCTTCCA and antisense: AAATGAGCCCCAGCCTTC; and *ACTB*, sense: ACTCTTCCAGCCTTCCTTCCT and antisense: CAGTGATCTCCTTCTGCATCCT).

### 2.5. Measurements of Cell Proliferation

Cell proliferation of RWPE-1, LNCaP, 22Rv1, and PC3 cell lines in response to EN2 treatment was determined using the Alamar blue fluorescent assay (Life Technologies, Madrid, Spain), as previously described [27,31]. Different concentrations of EN2 peptide (10^−6^ to 10^−9^ M) were initially tested in RWPE-1 and PC3 cell lines (Appendix A). Based on these results, the dose of 10^-7^ M was selected for further experiments in all the prostate cell lines included in subsequent experiments (i.e., RWPE-1, LNCaP, 22Rv1, and PC3 cells), as we found that this was the only dose that increase proliferation rate in PC-3 (but not RWPE-1) cells. Moreover, the increase in proliferation rate in response to 10^−7^ M of EN2 was corroborated in 22Rv1 cells (IGF1 and Paclitaxel were used as control of proliferation enhancement and inhibition, respectively). Briefly, cells were seeded in 96-well plates at a density of 3000 to 5000 cells per well and subsequently serum-starved for 24 h. Then, after 3 h of incubation with 10% Alamar blue serum-free medium, basal proliferation rate was obtained by measuring the fluorescent signal of reduced Alamar, exciting at 560 nm and reading at 590 nm using the FlexStation III system (Molecular Devices, Sunnyvale, CA, USA). Subsequently, medium was replaced by fresh medium containing 5% FBS and the treatments to be tested immediately after each measurement and proliferation rate was determined after 24 h incubation. Results were expressed as percentage referred to control (vehicle-treated). In all cases, cells were seeded per quadruplicate and all experiments were performed a minimum of three times.

### 2.6. Measurements of Migration Capacity

Cell migration was evaluated in RWPE-1 and PC3 cells by wound-healing technique as previously reported [27,28]. Briefly, 300,000 cells were cultured in 12-well plates and then a wound was made using a 200 µL sterile pipette tip on confluence conditions. Then, the wells were rinsed using PBS and subsequently incubated for 24 h in serum free medium. Wound healing was evaluated as the area of a rectangle centered in the picture 24 h after the wound vs. the area of the rectangle just after the wound was performed. At least three experiments were performed in independent days.

### 2.7. Measurement of Free Cytosolic Calcium Concentration ([Ca^2+^]_i_)

Cells were plated on coverslips at a density of 100,000 cells per well and changes in [Ca^2+^]_i_ in RWPE-1, LNCaP, and PC3 cell lines after treatment with EN2 protein were tracked in single cells using fura-2/AM (Molecular Probes, Eugene, OR, USA) as described previously [27,32].

### 2.8. Microarray of Gene Expression Profile

Microarray experiment was carried out using the Human Androgen Receptor Signaling Targets PCR Array PAHS-142Z (Qiagen, Limburg, Netherlands). Three independent passages of LNCaP cells treated for 24 h with EN2 protein mixed in one pool, and the respective vehicle-treated controls were used. Retrotranscription was performed using RT² First Strand Kit (Qiagen, Limburg, Netherlands) and expression was measured using RT² qPCR SYBR Green ROX (Qiagen, Limburg, Netherlands) in Stratagene Mx3000p system. Results were analyzed with Data Analysis Center (Qiagen, http://www.qiagen.com/shop/genes-and-pathways/data-analysis-center-overview-page/), following the manufacturer’s instructions.

### 2.9. Western Blot

RWPE-1, LNCaP, and PC3 cells were treated with EN2 peptide for 8 min for the evaluation of AKT, ERK, and Androgen Receptor (AR) phosphorylation levels by western blot using standard procedures, as previously reported [27,28,30]. Furthermore, AR and AR splice variants (SVs) phosphorylation levels by western blot were also evaluated in 22Rv1 cells. Briefly, proteins were extracted from cells seeded in 12-well plates using pre-warmed Sodium dodecyl sulfate- Dithiothreitol (SDS-DTT) buffer (62.5 mM Tris-HCl, 2% SDS, 20% glicerol, 100 mM DTT and 0.005% bromophenol blue), followed by sonication during 10 s and boiling for 5 min at 95 °C. Proteins were separated by SDS-PAGE and transferred to nitrocellulose membranes (Millipore, Billerica, MA, USA). Membranes were blocked with 5% non-fat dry milk in Tris-buffered saline/0.05% Tween 20 and incubated with the specific primary antibodies [p-AKT (Ser47; Ref. CS9271S), Akt (Ref. CS9272), p-ERK1/2 (Ref. CS4370), ERK1/2 (Ref. CS154) from Cell Signaling (Danvers, MA, USA); p-Androgen Receptor (p-AR; Ser210; Ref. AB71948) and AR (Ref. AB133273) for both full-length and SVs ARs, from Abcam (Cambridge, UK)], tubulin beta (TUBB) (Ref. #2128S, Cell-Signaling), and the appropriate secondary antibodies (Cell Signaling)]. Proteins were detected using an enhanced chemiluminescence detection system (GE Healthcare, Madrid, Spain) with dyed molecular weight markers. A densitometry analysis of the bands obtained was carried out with ImageJ software, using total protein as normalizing factor of correspondent phosphorylated protein or TUBB as normalizing factor of total proteins.

### 2.10. Determination of PSA and EN2 Levels by ELISA

PSA secretion was measured after EN2 treatment (10^−7^ M concentration) in the LNCaP cell line using a specific commercially available ELISA kit (DRG Diagnostics, Marburg, Germany). Briefly, cells were seeded in 12-well plates at 70% confluence in serum-starved medium and 24 h later media were collected and stored at −20 °C until measurement. Results are expressed as the percentage of PSA secretion vs. vehicle-treated cells. Three independent experiments were performed on separate days, in which cells were plated per triplicate. In addition, EN2 levels were determined in medium from RWPE-1, LNCaP, and PC3, as well as in urine from PCa patients and control individuals using a commercially available ELISA kit (Wuhan EIAAB Science Co., Wuhan, China) following the manufacturer’s instructions. All the information regarding specificity, detectability, and reproducibility for each of the assays can be accessed at the website of the company.

### 2.11. Statistical Analysis

Descriptive results were expressed as mean ± standard error of the mean (SEM) or median and interquartile range in case of quantitative data, and in absolute value and percentage in case of qualitative variable. Paired or unpaired (parametric or non-parametric) tests were performed to determine significant differences between two groups. The receiver operating characteristic (ROC) curve was performed for evaluation of the accuracy of EN2 as a diagnostic marker in the different tissues and fluids analyzed. *p*-values ≤ 0.05 were considered statistically significant and a trend for significance was indicated when *p*-values ranged between <0.1 and >0.05. Statistical analyses were performed using SPSS 17.0 (IBM SPSS Statistics Inc., Chicago, IL, USA) and GraphPad 7.0 (GraphPad Software, La Jolla, CA, USA).

## 3. Results

### 3.1. EN2 Is Overexpressed in Tissue and Urine Samples from PCa Patients Compared with Controls

Analysis of *EN1* and *EN2* mRNA levels in FFPE prostate pieces from patients with low-intermediate grade PCa revealed that *EN2* expression was significantly higher in tumor vs. non-tumor adjacent tissue, specifically in 24 out of 33 samples (72.73%; Figure 1a), whereas *EN1* mRNA levels did not differ (Appendix A). *EN2* overexpression was confirmed in an independent cohort of fresh PCa biopsies from high-risk/locally-advanced PCa patients (*n* = 23) vs. fresh normal prostate tissues (*n* = 7) (Figure 1b, left-panel). Of note, ROC analysis showed that *EN2* mRNA levels clearly discriminated between PCa and control subjects (AUC = 0.96; *p* < 0.001; Figure 1b, right-panel). These results were further corroborated by analyzing public databases obtained from The Cancer Genome Atlas (TCGA; *n* = 52 tumor and 52 non-tumor adjacent; Figure 1c) data portal, wherein 41 out of 52 samples (78.85%) showed a clear overexpression of EN2, and from the Memorial Sloan Kettering Cancer Center (MSKCC; *n* = 179) dataset (Figure 1d) which showed higher levels of *EN2* mRNA in PCa samples compared to controls. Although no other relevant clinical associations with Gleason score or other tumor-related pathologic parameters were found in these cohorts (data not shown), in silico analysis using the Grasso cohort indicated that EN2 expression tends to be overexpressed in CRPC samples (Appendix A).

In line with the previous data, mean urine EN2 protein levels were clearly elevated in PCa patients compared with healthy controls (Figure 1e, left-panel). Specifically, EN2 was detected in urine samples from 18 out of 24 (75%) of the patients with PCa vs. only in 45% of controls. In this sense, ROC curve analysis suggested the potential of urine EN2 levels to discriminate between PCa patients and controls [AUC = 0.66 (0.50–0.83); *p* = 0.06] (Figure 1e, right-panel).

### 3.2. EN2 Is Overexpressed, Secreted, and Modulates Aggressiveness Features in PCa Cells

Analysis of the expression of *EN2* in different prostate cell lines (Figure 2a) revealed that the mRNA of this variant was virtually absent in the normal-like prostate cell line RWPE-1, while it was clearly over-expressed in the two PCa cell lines analyzed, LNCaP and PC3. Available online data from the Cancer Cell Line Encyclopedia further confirmed this overexpression in PCa cell lines compared to normal prostate cells (Appendix A). Consistently, EN2 protein was found to be secreted from PCa cell lines (determined by ELISA in medium), while its levels were under the detection limit in the normal RWPE-1 cells (Figure 2b). In support of this latter observation, our data also suggest that the EN2 present in the urine might be mainly derived from prostate cells because urine EN2 levels were clearly increased after DRE in the same cohort of PCa patients previously described in Table 3 (*n* = 24; Appendix A).

Treatment with EN2 protein significantly increased cell proliferation rates in all PCa cell lines analyzed (LNCaP, 22Rv1 and PC3) but not in the normal RWPE-1 cells, compared to vehicle-treated controls (Figure 2c). In contrast, treatment with EN2 protein increased migration rate in both RWPE-1 and PC3 cell lines (Figure 2d). Interestingly, PSA secretion was also augmented after 24 h of treatment with EN2 protein in LNCaP cells (Figure 2e).

### 3.3. EN2 Modulates Key Signaling Pathways and Molecular Targets in PCa Cells

In order to unveil the molecular mechanisms underlying the pro-tumorigenic actions of *EN2* in PCa, we first determined the capacity of EN2 protein to modulate free cytosolic calcium concentration ([Ca^2+^]_i_). Our results revealed that treatment with EN2 protein did not alter [Ca^2+^]_i_ kinetics in RWPE-1, LNCaP, or PC3 cells (Appendix A), while ionomycin elicited the appropriate response in all cell lines, indicating that EN2 does not alter this signaling pathway. In marked contrast, western blot analysis revealed that treatment with EN2 protein (10^−7^ M; 8 min) increased the phosphorylation of AKT and AR, but not ERK protein in LnCaP cells (Figure 3 and Appendix A). Furthermore, AR signaling modulation by EN2 was also corroborated in the androgen sensitive 22Rv1 cells, wherein treatment with EN2 increased the phosphorylation of full-length (Figure 3 and Appendix A) but also the splicing variants of AR (Appendix A), while it did not alter total AR levels. On the other hand, no changes in the phosphorylation levels of these proteins were found in response to EN2 treatment in normal-like RPWE-1 or androgen-insensitive PC3 cells (Figure 3 and Appendix A).

In addition, to explore a putative association between EN2 and key factors in PCa, we measured, in LNCaP cells treated with EN2, a PCR Array of human androgen receptor signaling pathways, which allows the measurement of the mRNA levels of a wide number of key genes involved in pathways related with AR. This array revealed that several genes, mainly related with tumor progression, were altered when cells were treated with EN2 compared with vehicle-treated control cells (Figure 4a). Specifically, when considering significant differences in genes with a fold change higher than 2, the array revealed the upregulation of *EGR3* and *PTGS1* and the downregulation of *GSTP1* in response to EN2 protein treatment (Figure 4b).

## 4. Discussion

The high incidence and prevalence of PCa represent a major health problem worldwide [1], whose management is also hampered by the limited availability of appropriate diagnostic, prognostic, and therapeutic tools. In particular, PCa screening based on the current gold-standard PSA and the DRE remains controversial, mainly due to the high rate of over-diagnosis and unnecessary prostate biopsies [33]. In addition, although the molecular characterization of this type of cancer has provided new predictive and prognostic markers, as well as novel therapeutic targets [7,34], their universality and applicability are still a matter of debate. For these reasons, it seems crucial to identify new molecular biomarkers which would help to refine the diagnosis, to improve the prediction of the prognosis and behavior of the tumor, and to provide tools to develop novel therapeutic approaches.

In this scenario, earlier studies suggested that the gene family of homeodomain transcription factors might play a relevant role in the pathophysiology of PCa and, hence, that they could provide novel tools for the diagnosis, prognosis, and/or treatment of this pathology [21,35,36,37]. Specifically, previous reports showed that *EN2* is overexpressed in tumor prostate tissues and PCa cell lines compared with normal prostate tissue and normal-like cell lines [19,20]. In the present study, we have confirmed and expanded these findings by corroborating in independent and ample cohorts of patients that *EN2* is overexpressed in PCa samples with different grades of differentiation and aggressiveness, in comparison with normal prostate tissues but also with their respective adjacent non-tumor tissue. It is methodologically worth noting that the differences could be observed in both FFPE and frozen samples. Moreover, these results have been further substantiated by analyzing public databases from The Cancer Genome Atlas data portal and from the MSKCC dataset, which indicate that *EN2* is overexpressed in different cohorts of PCa patients and; therefore, suggest the universality of this biomarker.

Most relevant from the diagnostic point of view is the fact that EN2 can be found in urine, the less invasive liquid biopsy, wherein its levels have been shown to be elevated in PCa patients compared to controls [20,21,22,23,24]. In particular, Morgan et al. demonstrated that patients with PCa had 10-fold increased urine EN2 levels compared to controls, showing a specificity of 88% to diagnose PCa [20], which suggested the putative utility of urine EN2 levels as a novel non-invasive PCa biomarker. Indeed, the same group validated these results by using patients with high risk of PCa included in the IMPACT cohort [21], and later correlated the urinary levels of EN2 with tumor stage and volume in patients treated with radical prostatectomy (first-pass and midstream urine samples were evaluated) [22,23]. However, a subsequent study by Marszałł et al. did not find different urinary levels of EN2 between patients with and without PCa when using complete urine samples, although they found higher EN2 urinary levels in urine after prostate massage [24]. Consequently, in that the sensitivity and specificity of this biomarker is considerably variable among studies and the values can fluctuate depending on the existence of previous prostate massage, the appropriateness and accuracy of EN2 as a PCa diagnostic biomarker, as well as the methodological procedure for EN2 assessment in urine samples, are still a matter of debate [20,21,22,23,24]. In this scenario, our results are in accordance with Morgan et al., showing higher levels and discriminatory capacity of EN2 in urine from patients with PCa versus controls, using an independent cohort of patients. Nevertheless, it should be taken into account as a limitation that the use of TRUS biopsy for PCa diagnosis, despite being the current standard in most populations, suffers from some false negative results and random error compared with template biopsy. In this series of studies, the authors recommended measuring the first part of the urine, but they have also found association with PCa volume with EN2 mid-urine levels. In contrast, we have observed similar differences by measuring EN2 levels in whole urine, which may suggest that EN2 urinary levels could represent a valuable PCa diagnostic tool without the necessesity of prostate stimulation if whole urine is used. However, this hypothesis should be further validated in subsequent studies using different, ampler cohorts of patients. It is also important to note that, as Marszałł et al. [24], we have analyzed EN2 urinary levels with a commercial ELISA kit and, maybe, our results could be improved with better EN2 detection systems. Indeed, Morgan et al., who first described the possible utility of this marker, used an non-commercial ELISA but their method has not been validated by other groups [20].

Interestingly, earlier studies also suggested that EN2 could play a tumorigenic role in PCa in that its silencing is associated to a decrease in PCa cell proliferation [19]. One of the most striking features of EN2 is that its protein does not seem to be localized in the nucleus of PCa cells but, rather, close to the luminal border of the cells, associated to secretory blebs [20]. This is, indeed, consistent with the observation that cells from different established PCa cell lines can release EN2 protein to the medium (data presented herein and in Morgan et al. [20]), and that urine EN2 levels increase after DRE in PCa patients (data presented herein and in Marszałł et al. [24]). Altogether, these results suggest that secreted EN2 could play a pathological role in PCa that remains poorly known. These reasons prompted us to explore herein the putative tumorigenic role of secreted EN2 protein in normal and tumor prostate cells, inasmuch as this information could pave the way towards the identification and development of novel therapeutic avenues in PCa. This approach led us to demonstrate, for the first time, that secreted EN2 protein can act on normal and tumor prostate cells by modulating certain signaling pathways and cancer-associated genes, which ultimately results in an enhance tumorigenic potential in these cells (i.e., an increased capacity to proliferate, migrate, or secrete PSA). In particular, treatment with exogenous EN2 protein elicited an increase in the proliferation capacity of the PCa cell lines LNCaP and PC3, an increase in the capacity to migrate of normal-like RWPE-1 and PC3 PCa cells and an increase in PSA secretion from LNCaP cells, which are, all of them, parameters directly associated to the tumorigenic capacity of these cells [38]. Interestingly, our results also show that EN2 treatment evoked a modest but significant increase in the phosphorylation rate of full-length AR in LNCaP and full-length and SVs AR in 22Rv1, as well as an increase in phosphorylation rate of AKT in LNCaP. In this sense, the PI3K/AKT signaling pathway is a frequently dysregulated pathway in cancer [39], and specifically in PCa [40], wherein signaling cross-talk and functional synergism between PI3K/AKT and AR pathways have been previously reported [41]. Furthermore, the association between the dysregulation of full-length and SVs AR signaling and the process of promoting oncogenesis of all stages of PCa has been also widely demonstrated [42]. In support of these findings, our present results indicate that this tumorigenic capacity could likely be associated to the modulation of the expression of certain cancer-associated genes, in that the PCR-based array implemented herein to analyze the response of LNCaP cells to EN2 treatment revealed a relevant modulation of a discrete number of genes, including the upregulation of *PSTGS1* and *EGR3* and the downregulation of *GSTP1*. It is worth noting that both upregulated genes have been clearly shown to be involved in the association between inflammation and cancer [43,44]. In particular, EGR3 has been previously reported to be overexpressed in PCa cells and to upregulate inflammatory cytokines such as IL6 and IL8, which play an important role in PCa and contribute to disease progression and to the onset of castration resistance [43,45]. We also found that EN2 treatment elicited a significant down-regulation of *GSTP1*, which is known to be hypermethylated in PCa and to be correlated with the aggressiveness of the disease [46]. Nevertheless, since we observed that EN2 treatment had also the capability of modulating some functional parameters in RWPE-1 and PC3 cells, wherein it did not alter the phosphorylation levels of AKT, we could conclude that additional pathways activated by exogenous EN2 may exist, which may help to explain the tumorigenic role exhibited in these cell lines.

When viewed together, our results provide compelling evidence to support the potential value of EN2 as a non-invasive diagnostic biomarker for PCa, and offer, as well, novel valuable information to consider its putative utility to develop new therapeutic tools in this pathology. In particular, we expanded and validated the higher expression of *EN2* in PCa tissue vs. normal prostate, as well as its elevated levels in urine samples from PCa patients. In addition, we demonstrate herein, for the first time, that secreted EN2 protein could act as a tumorigenic factor in normal and tumor prostate cells, by modulating key functional parameters and signaling pathways. Therefore, these data invite to explore further the identification and development of novel therapeutic targets related to *EN2* in this high incidence and prevalent pathology.

## Figures and Tables

**Figure 1 jcm-08-01400-f001:**
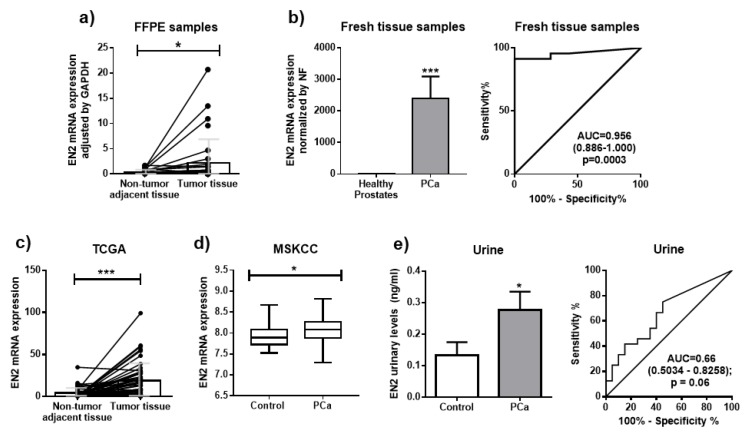
*EN2* is overexpressed in prostate tissue and urine from PCa patients. (**a**) Paired analysis of EN2 mRNA expression levels in PCa tissue and matched adjacent non-tumor tissue from 33 Formalin-fixed paraffin-embedded (FFPE) prostatectomy samples. Absolute mRNA levels were determined by qPCR and adjusted by *GAPDH* housekeeping gene. (**b**) mRNA expression levels of *EN2* in a battery of 23 PCa samples and compared to the expression levels found in seven normal prostates. Absolute mRNA levels were determined by qPCR and adjusted by normalization factor (NF). Receiver operating characteristic (ROC) curve analysis to determine the accuracy of EN2 to discriminate between tumor and healthy tissue. (**c**) Analysis of *EN2* mRNA expression levels in 52 PCa samples and 52 non-tumor adjacent samples from TGCA data set. (**d**) Analysis of *EN2* mRNA expression levels in 29 non-tumor tissue and 150 PCa tissues from the MSKCC data set. (**e**) Evaluation of EN2 levels as a non-invasive PCa diagnostic marker. EN2 urinary levels in 24 PCa patients (filled bars) compared to 20 controls (healthy and negative biopsy patients; open bars), determined by ELISA assay, without prostate massage (left panel). ROC curve analysis to determine the accuracy of EN2 to discriminate between tumor and healthy patients (right panel). Data represent mean ± SEM. * *p* ≤ 0.05, *** *p* < 0.001 indicate values that significantly differ between groups.

**Figure 2 jcm-08-01400-f002:**
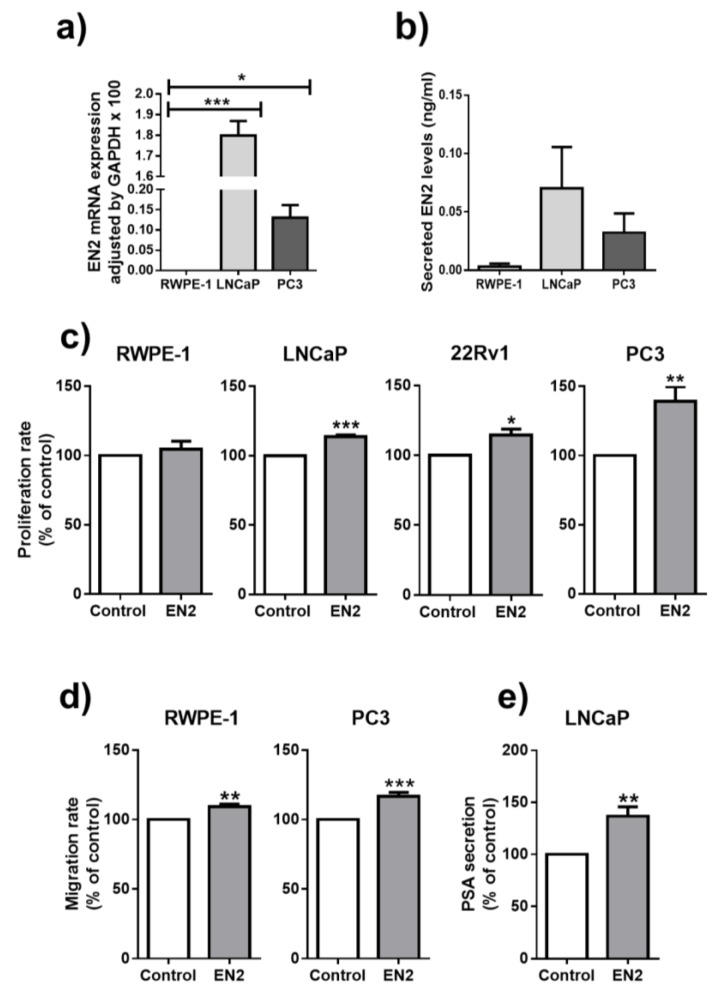
*EN2* expression and its functional role in prostate-derived cell lines. (**a**) *EN2* mRNA expression levels in normal-like prostate cell line, RWPE-1, and PCa cell lines, LNCaP and PC3, determined by qPCR and adjusted by glyceraldehyde-3-phosphate dehydrogenase (GAPDH) levels. (**b**) EN2 secretion levels from RWPE-1, LNCaP, and PC3 cell lines, determined by ELISA. (**c**) Effect of 24 h treatment with EN2 protein on cell proliferation rate in (from left to right) RWPE-1, LNCaP, 22Rv1, and PC3 cell lines compared to vehicle-treated controls. (**d**) Effect of 24 h treatment with EN2 protein on cell migration rate in RWPE-1 and PC3 cell lines, compared to vehicle-treated control. (**e**) PSA secretion from LNCaP cell line treated with EN2 protein compared with vehicle-treated controls (after 24 h culture) determined by a specific ELISA kit. Data represent mean ± SEM and they are expressed as percentage of vehicle-treated controls (set at 100%) within experiment. * *p* ≤ 0.05, ** *p* < 0.01, *** *p* < 0.001 indicate significant differences compared to control.

**Figure 3 jcm-08-01400-f003:**
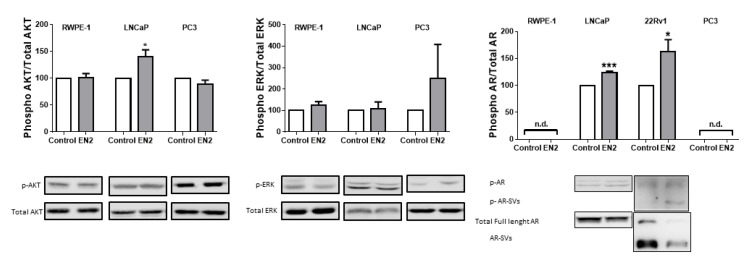
Downstream consequences of EN2 treatment in RWPE-1, LNCaP, 22Rv1, and PC3 cells. Phosphorylation of key signaling pathways (AKT, ERK, and AR; from left to right) after EN2 treatment during 8 min, compared with non-treated control (full western blot images in Appendix A). n.d means non-detectable levels. AR-SVs = AR splice variants. Data represent mean ± SEM and they are expressed as percentage of the ratio (set at 100%). * *p* ≤ 0.05, *** *p* < 0.001 indicate significant differences compared to control.

**Figure 4 jcm-08-01400-f004:**
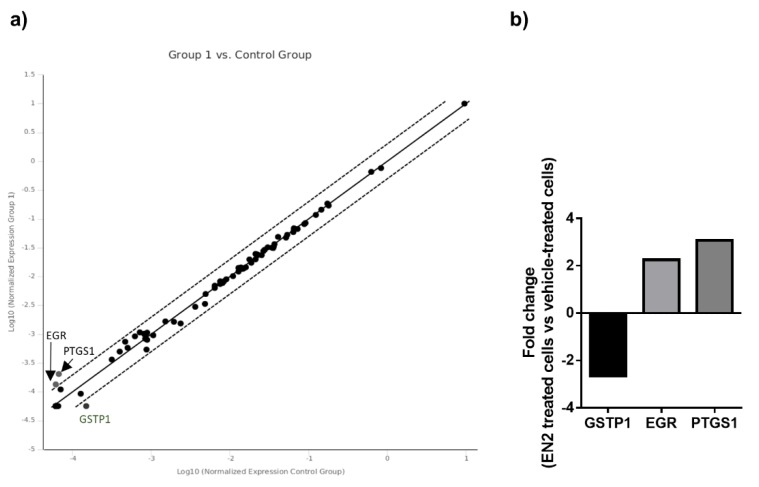
PCR array of human AR signaling pathway. (**a**) Representation of differences (two-fold change) between control and EN2-treated LNCaP cells using scatter plot. Upregulated genes are at the top of the image and downregulated genes at the bottom. (**b**) Representation of log_2_ fold changes in significantly-altered genes [Glutathione S-transferase P 1 (*GSTP1*), Early growth response 3 (*EGR3*), and Prostaglandin-Endoperoxide Synthase 1 (*PTGS1*)] between control and EN2-treated LNCaP cells.

**Table 1 jcm-08-01400-t001:** Overall clinical and demographic data and expression levels measured in formalin-fixed paraffin-embedded (FFPE) prostate pieces from patients with clinically-localized prostate cancer (PCa).

Variable	Overall
Number of patients	33
Age at diagnosis	
Median (IQR)	62 (58–66)
BMI	
Median (IQR)	27.7 (25.8–31.3)
PSA level, ng/mL	
Median (IQR)	6 (4.4–9.5)
Gleason score in prostatectomy specimen (%)	
6	9 (27.3)
7	23 (69.7)
8	1 (3)
EE, *n*° (%)	21 (63.6)
PI, *n*° (%)	28 (84.8)
VI, *n*° (%)	8 (24.2)
Relative *EN2* mRNA expression in FFPE piece ^a^	
Tumor tissue	
Median (IQR)	0.173 (0.002–1.473)
Non-tumor adjacent tissue	
Median (IQR)	0.008 (0.000–0.477)
Ratio tumor/non-tumor tissue	
Median (IQR)	3.451 (1.260–12.212)
Relative *EN1* mRNA expression in FFPE piece *	
Tumor tissue	
Median (IQR)	0.723 (0.209–2.828)
Non-tumor adjacent tissue	
Median (IQR)	0.421 (0.149–1.145)
Ratio tumor/non-tumor tissue	
Median (IQR)	1.197 (0.325–3.249)

EE = extraprostatic extension; PI = perineural invasion; VI = vascular invasion; FFPE = formalin-fixed paraffin-embedded; IQR = interquartile range. a FFPE prostate piece with delimited tumor tissue and non-tumor adjacent tissue. * EN1 (n = 18) and EN2 expression (Ct) was calculated by qPCR, adjusted with glyceraldehyde-3-phosphate dehydrogenase (GAPDH) and analyzed by Delta (Ct) method.

**Table 2 jcm-08-01400-t002:** Overall clinical and demographic data of fresh samples from patients with normal prostates and PCa samples.

Variable	Overall	Control	PCa
Patients	30	7	23
Age at diagnosis			
Median (IQR)	73 (64–79)	67 (59–79)	76 (67.0–80.0)
PSA level, ng/mL			
Median (IQR)	-	-	40 (22–70)
Dyslipidemia (%)	7 (23.3)	2 (28.6)	5 (21.7)
Diabetes (%)	8 (26.7)	2 (28.6)	6 (26.1)
* BMI			
Median (IQR)	27.19 (25.2–29.83)	25.87 (24.50–34.24)	27.44 (25.46–29.65)
Gleason score			
=7	-	-	8 (34.8)
>7	-	-	15 (65.2)
EE (%)	-	-	6 (26.1)
PI (%)	-	-	14 (60.9)
^#^ Metastasis (%)	-	-	13 (56.5)
N° samples (%) in whichEN2 was detected	25 (83.3)	3 (42.9)	22 (95.7)
ª Median(IQR) *EN2* mRNAexpression	445 (8–2265)	0 (0–9)	874 (173–2650)

PCa = prostate cancer; EE = extraprostatic extension; PI = perineural invasion. ^#^ Metastasis (N or M stage) **ª**
*EN2* expression (Ct) was calculated by qPCR, adjusted by normalization factor (Beta-actin (ACTB) and GAPDH) and analyzed with copy numbers using a standard curve. * BMI; *n* = 22 (missing data).

**Table 3 jcm-08-01400-t003:** Demographic and clinical characteristic of patients included in the study of urine EN2 levels.

Variable	Control (*n* = 10)	Negative Biopsy (*n* = 10)	PCa (*n* = 24)	*p-*Value
Age, Years				
Median (IQR)	56 (52–61)	56 (53–59)	68 (59–71)	<0.01
Waist circumference, cm				
Median (IQR)	107 (99.5–111.8)	103 (98.3–109)	104.5 (100–111.5)	0.88
BMI				
Median (IQR)	32.4 (27.5–33.1)	30.1 (27.33–31.97)	29 (26.67–30.48)	0.15
PSA, ng/mL				
Median (IQR)	0.87 (0.6–1.6)	3.6 (3.0–4.0)	5.7 (4.6–9.7)	<0.01
DRE, abnormal (%)	0 (0)	0 (0)	9 (37.5)	<0.01
Prostate Vol, cc				
Median (IQR)	-	37 (23.1–45.8)	37 (25.59–52.5)	0.55
1° Biopsy (%)	-	9 (90)	17 (70.8)	
Gleason grade (%)				
=6			6 (25)	
≥7			18 (75)	
Nº Pathologic cores				
Median (IQR)			2 (1–4)	
EN2 Urine				
Median (IQR)	0 (0–0.21)	0.02 (0.00–0.30)	0.19 (0.01–0.43)	0.05

PCa = prostate cancer; Yrs = year; cm = centimeters; BMI = body mass index; IQR = interquartile range; PSA = Prostate specific antigen; DRE = digital rectal examination; Vol = volume. Statistical analysis: Non-parametric test for independent groups comparing non-tumor (control + negative biopsy) versus PCa patients.

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
