# Peer review of "Oncogenic Role of Secreted Engrailed Homeobox 2 (EN2) in Prostate Cancer"

_jcm, 2019, doi:10.3390/jcm8091400_

Round 1

Reviewer 1 Report

This manuscript should be ready for publication!

Author Response

We sincerely thank the reviewer for supporting our work.

Reviewer 2 Report

The reviewer needs two more response as the following:

1) The authors should explain why total AR level decreased after EN2 treatment as the author can see in figure 3.

2)Did authors check PSA expression after EN2 treatment in 22Rv1 cells? 

Author Response

This manuscript is a resubmission of an earlier submission. The following is a list of the peer review reports and author responses from that submission.

Round 1

Reviewer 1 Report

This is a well written manuscript using reasonably sound methodology and intepretation of results.It scores low on originality as it corroborates previous studies, and adds a little up to date informatics. The mRNA studies should have been supported by western blots to show translation, the numbers of individuals used in the urinary studies are low. The authors did not explain helathy controls- the occult rate of prostate cancer is around 7% in the general population (ESPRC study). No mention of tumour bulk or Gleason in the tissue correlations. The conclusions were 

reasonable. 

Reviewer 2 Report

The authors evaluated EN2 expression levels between prostate cancer cells and benign ones by targeting both cell lines and tissues. Moreover, they showed EN2 levels increased in urine of prostate cancer patients in comparison to that of ones without prostate cancer. They also showed biological roles of EN2 in prostate cancer progression.

The reviewer thinks that there are several flaws which should be revised and solved for acceptance to the journal.

Did the EN2 in urine really derived from prostate cancers themselves? The authors should present EN2 levels in urine before and after rectal examination in order to clearly show source of EN2 in urine. 22Rv1 expresses AR splicing variant basically. In Figure 3 how did you calculate pAR/ total AR ratio? Please show clearer WB figure both full length AR and splicing variant AR together. The authors used AR signaling pathway PCR array to evaluate the effect of EN2 in LNCaP cells. Please showed the results of KLK3 (PSA) and other AR regulated genes such as NDRG1, which might change their expression. In discussion, the authors describe that “EN2 may activate, at least in LNCaP cells, the PI3K /AKT pathway in an AR dependent manner “, in line 409-410. However, the authors did not present any direct data with regards to PI3K/AKT pathway and AR. Thus, the reviewer recommends changing the description more weakly about the relationship. In Figure 1 a and c “adyacent” should be “ adjacent”.

Reviewer 3 Report

In this work, the authors have investigated the expression status of EN2 in prostate tissues and cell lines. They have also investigated the possible functional role of EN2 in promoting the malignant progression of the cancer cells. The study has also included some work on how EN2 played its functional role and the possibility of using EN2 as a therapeutic target has also been assessed. This study involved in a relatively large amount of experimental work and study has been conducted in a standard manner. Most of the experimental work has been designed properly and some of data generated from this study is potentially valuable.

One of the concerns in the tissues work is the control samples. Adjacent tissues to carcinoma cannot be regarded as normal. Some changes may have already occurred. The proper control is benign prostatic tissues or tissues from subjects who had not been diagnosed any prostatic diseases.  The authors should also investigate whether the increase in EN2 level is associated with the increasing Gleason scores of the cases or whether is significantly associated with a shorter survival of the patients. The association of the increased EN2 expression with some malignant characteristics of the cells is observed only in part of the cell lines and therefore the conclusion is unconvincing. A migration assay and an anchorage-independent growth assay should be employed to test the cell lines. If possible, animal work should be performed to confirm that EN2 expression can really promote the malignant progression in part (such as androgen-responsive cells) or in the entire prostatic cells.
